# Green Supplier Selection in an Uncertain Environment in Agriculture Using a Hybrid MCDM Model: Z-Numbers–Fuzzy LMAW–Fuzzy CRADIS Model

**Adis Puška** [1,*]![iD], **Darko Božanić** [2]![iD], **Miroslav Nedeljković** [3] **and Miljojko Janošević** [2]

1    Faculty of Agriculture, Bijeljina University, Pavlovića put bb, 76300 Bijeljina, Bosnia and Herzegovina
2    Military Academy, University of Defence in Belgrade, Veljka Lukica Kurjaka 33, 11000 Belgrade, Serbia
3    Institute of Agricultural Economics, Volgina 15, 11060 Belgrade, Serbia
*    Correspondence: adispuska@yahoo.com

**Abstract:** The goal of this research was to find a selection of green suppliers (GSS) that will, in the best way, help agricultural producers to apply green agricultural production using uncertainty in decision making. In order to avoid the possibility of uncertainty in the expert decision making, Z-numbers were used together with the fuzzy LMAW (Logarithm Methodology of Additive Weights) method and fuzzy CRADIS (Compromise Ranking of Alternatives from Distance to Ideal Solution) method. By applying Z-numbers and the fuzzy LMAW method, the weighting coefficients of the criteria were determined, where the experts, in addition to the criteria ratings, also defined the degrees of certainty in the criteria ratings they gave. The obtained results indicated that the criteria related to price and qualities are the most important during the selection process. To select the best alternative, the CRADIS method modified with Z-numbers and fuzzy numbers was applied. The results obtained by applying this method showed that suppliers A2 and A3 have the best characteristics and are the first choice for the procurement of raw and production materials. As part of the paper, the validation of the results and the sensitivity analysis of the model were carried out by conducting the procedure of comparing the obtained results with the results obtained by other MCDM methods and changing the weighting coefficients of the criteria. These analyses indicated that the model presented provides stable results. The conducted research showed how Z-numbers can be used to reduce uncertainty in decision making and how Z-numbers can be used with other fuzzy methods to perform GSS.

**Keywords:** green supplier selection; Z-numbers; agricultural production; fuzzy LMAW method; fuzzy CRADIS method

**MSC:** 90B50

## 1. Introduction

Changes in the market are partly caused by changes in consumer behaviour. Consumers are now looking for efficient and environmentally friendly products, especially when it comes to food [1]. That is why there is an increasing pressure on food producers to produce as much as possible while making the food environmentally acceptable. In addition, the pressure is greater due to legal regulations and competitors [2] and not only customer demands. The production of ecologically acceptable food has become a priority for agricultural producers. In order for this food to be recognizable, a special labelling of this food is used [3]. Consumer demands are mostly about the use of ecologically acceptable products, and this has led manufacturers to adapt to these changes, starting from the procurement of raw and production materials [4]. All of these changes in the market can affect the entire supply chain of agricultural and food products [1]. Accordingly, manufacturers should seek green supply chain management strategies to respond to market demands [5].

It is difficult to produce environmentally friendly food without suitable raw and production materials. Therefore, the first step in the production of ecologically acceptable food is the choice of suppliers of ecological raw and production materials. Since these are procured from a supplier, it is necessary to choose a supplier that respects ecological principles in its business. In this way, it is necessary to choose a green supplier. Green supplier selection (GSS) implies primarily the development of an efficient ecological supply chain [6]. A key tool for the implementation of GSS is the inclusion of environmental criteria in the selection of suppliers [7]. On that occasion, economic and environmental criteria are used to select suppliers [8]. Economic criteria are used so that the raw and production materials procured from suppliers are at favourable prices and of an enviable level of quality. The ecological criteria in this selection serve to ensure that the raw and production materials that are procured meet the set ecological standards and help manufacturers to apply green supply chain management (GSCM) [9]. That is why these criteria are used in GSS.

Suppliers selected using GSS help manufacturers to reduce their negative environmental impact and maximize economic performance [10]. GSS helps producers with the existence of mass information and possible risks of collected data [5]. However, GSS represents a laborious task for the manufacturer that includes many challenges that arise, all the way from the evaluation to the final selection [11]. Choosing the right supplier can ensure lower operating costs and better quality products with a minimal impact on the environment [12].

GSS is considered a problem that is solved by applying group multicriteria decision making [13]. This choice is made by applying different criteria with regard to solving environmental problems during GSS, and these criteria can be presented qualitatively and quantitatively [14]. Qualitative criteria are advised when one wants to explain quality, efficiency, etc., where the linguistic assessment of the criteria is used. The application of linguistic values entails the application of fuzzy logic in GSS. However, not all information can always be available, so it is necessary to make decisions with a certain amount of uncertainty. In order to solve this problem, Z-numbers will be used in this paper.

Unlike classic fuzzy numbers, Z-numbers provide wider possibilities and are used to consider uncertainty in decision making [15]. The concept of applying Z-numbers was proposed by Professor Zadah [16]. Since his paper, the concept of Z-numbers has seen many applications, including in the selection of suppliers. Duan et al. [17] applied the concept of Z-numbers to allocate orders through GSS and to save costs and improve performance and competitive advantage. Li et al. [18] used Z-numbers and relative probability theory to select a supplier using the Z-TOPSIS (Z-Technique for Order of Preference by Similarity to Ideal Solution) method. Hosseini et al. [19] used Z-numbers to address potential uncertainty and ambiguity in the selection of sustainable suppliers. Fan et al. [20] used the Z-MABAC (Z-Multi-Attributive Border Approximation Area Comparison) method to select third-party logistics providers. Forghani et al. [21] used principal component analysis (PCA) to reduce the number of criteria for supplier selection, while supplier selection was performed using the Z-TOPSIS method. Kang et al. [22] performed supplier selection using a genetic algorithm (GA) to determine criteria weights and Z-numbers. Aboutorab et al. [23] applied the BWM (Best Worst Method) with Z-numbers to perform supplier selection. Liu et al. [24] performed supplier selection using the ANP (Analytic Network Process) methods for determining the weight of criteria and TODIM (Tomada de Decisão Iterativa Multicritério) on the example of nuclear energy production.

In these and other examples, it can be seen that Z-numbers were successfully used in the example of supplier selection. The specificity of this paper is that buyers of agricultural products have become increasingly demanding, and producers must adapt to them. In order to achieve this, they purchase raw and production materials from certain agricultural companies and pharmacies. Agricultural companies for the sale of raw and production materials in Bosnia and Herzegovina (BiH) play a major role for agriculturalists [25]. That is why it is necessary to carry out GSS on the example of an agricultural company because

it supplies agricultural producers. Based on that, the goal of this paper is to perform GSS on the example of a distribution agricultural company in order to acquire raw and production materials that are necessary for green agricultural production. Since agricultural pharmacies using GSS do not have all the necessary information about suppliers, decision making based on Z-numbers will be applied to reduce uncertainty. In addition, this paper will answer the following questions:

- How do we improve green agricultural production by applying GSS?
- How do we reduce uncertainty in decision making by applying Z-numbers?
- How do we apply expert decision making in GSS?

In order to answer these questions, this paper will use the methods of multicriteria analysis (MCDA), namely: fuzzy LMAW (Logarithm Methodology of Additive Weights) method and fuzzy CRADIS (Compromise Ranking of Alternatives from Distance to Ideal Solution) method. The fuzzy LMAW method will determine the weighting coefficients for the used criteria, while the fuzzy CRADIS method will determine the ranking for the selected suppliers. Of course, before implementing these methods, it is necessary to transform Z-numbers into classical fuzzy numbers.

The contribution of this paper is twofold. First of all, the GSS problem is solved by applying a scientifically recognized methodology. On the other hand, there is also a significant methodological contribution, because the LMAW and CRADIS methods modified with Z-numbers and fuzzy numbers are presented for the first time in the paper. The third use of Z-numbers contributes to stability in decision making because uncertainty is included in decision making because decision makers do not always have complete information.

Apart from the introduction, this paper is divided into the following sections. In the Section 2, the methodology and methods that will be used with the GSS source will be presented. The Section 3 of this paper will show the application of these methods, and GSS will be performed on a practical example, followed by the presentation of the obtained results. In the Section 4, the obtained results will be analysed through discussion, while the Section 5 presents the most important results of this research.

## 2. Methodology and Methods

The methodology used in this paper consists of three phases:

- Phase 1—The initial phase of research
- Phase 2—Research results
- Phase 3—Evaluation of results and sensitivity analysis

The first phase of this research includes defining the goal of the research, which is presented in the introduction. After that, it is necessary to determine the example from practice on which the research will be carried out. As mentioned, the research will be conducted on the example of the agricultural company "Agromarket" in Bijeljina. This company was chosen because it has three distribution centres located in Bijeljina, Laktaši and Sarajevo. In addition, this company is becoming an increasingly important participant in the agricultural markets of Bosnia and Herzegovina. The main activity of this agricultural company is the trade and distribution of raw materials for agriculture, garden and tool programs, and agricultural tools and machines. It cooperates with agricultural producers and supplies them with the necessary raw and production materials. This is why it is a better choice for determining a green supplier than individual agricultural producers. In order to carry out GSS, experts were chosen to assess the producers of raw and production materials needed for green agricultural production. Three experts with multiple years of experience in the procurement of raw and production materials were selected: two are agricultural engineers, and one is an expert technology engineer in the field of food production.

Along with them, the criteria with which GSS was to be carried out were determined (Table 1). About twenty criteria were presented to the experts, but they chose 10 criteria, which, in their opinion, are the most important for GSS in agriculture. Out of the 10 selected criteria, five criteria are those that concern the characteristics of ecological suppliers and

five criteria that concern the economic characteristics of suppliers. In this way, the same importance is given to both ecological and economic features because the same number of criteria are taken. After selecting the criteria, the experts identified six suppliers from whom they procure raw and production materials.

**Table 1.** Criteria for GSS.

| Id | Name | Description | References |
|----|------|-------------|------------|
| C1 | Green product | Products that are environmentally friendly | [2,4,10,11,19,26] |
| C2 | Eco design of the product | Designing in accordance with ecological standards | [2,10,11,19,25,27,28] |
| C3 | Environmental management system | Application of ISO 14,001 standard | [2,5–7,9–11,17,21,25,27,28] |
| C4 | Recycling | Material and waste reuse | [4,17,19,25,26,28] |
| C5 | Pollution control | Standards that reduce environmental impact | [2,4,10,11,14,25] |
| C6 | Quality | Degree of satisfying customer requirements | [2,4–6,9–11,14,17,19,21,27] |
| C7 | The price | Monetary value | [2,3,6,9,17,25,28] |
| C8 | Logistics service | Supply from supplier to customer | [4,14,17,19,20,25,27] |
| C9 | Innovativeness | Application of new products | [4,10,21,25] |
| C10 | Technological capacities | Technological capacities of suppliers | [2,4,6,9,10,14,17,20,25] |

Based on the selected criteria and alternatives, a research questionnaire was created. Questionnaires were sent to the experts, where they determined the importance of the criteria and evaluated the selected suppliers with the determined criteria. When evaluating the importance of the criteria, the experts determined this using a value scale of nine degrees ranging from Absolutely low (AL) to Absolutely high (AH). Each expert gave an assessment that ranged in this interval, depending on how important, in their opinion, that criterion was for the choice of supplier. After determining the value of the criteria, it was necessary to evaluate the supplier. The second part of the questionnaire was responsible for evaluating the supplier, which was conducted in the following way. Each provider was rated on a nine-point linguistic scale ranging from Absolutely low (AL) to Absolutely high (AH). The experts assessed how well, according to them, the particular supplier meets the set criteria. However, experts did not have all the information about suppliers, so there was a certain level of uncertainty. In addition to evaluating the suppliers, the experts also determined the level of uncertainty with a linguistic scale of values up to five degrees ranging from Very small (VS) to Very high (VH). In this way, the evaluation of the selected suppliers was carried out using Z-numbers.

The next step of the methodology was the application of the second phase of the research and the research results. After the data were collected from the experts, the results of the research were calculated. First, the weight of the criteria was calculated and then it was determined which of the suppliers best meets the objectives of this research. The fuzzy LMAW method was used when calculating the weight of the criteria. The goal of this method was to determine the weight value of the criteria based on the distance of the alternatives from the reference point [29]. After the weights of the criteria were determined, the suppliers were ranked based on the experts' evaluations. The ranking of suppliers was determined using the fuzzy CRADIS method. This method ranks alternatives by calculating the deviation of alternatives from ideal and anti-ideal solutions [30]. Both of these methods were used in the form of Z-numbers.

The next step of the methodology was the application of the third phase of the research, namely the evaluation of results and sensitivity analysis. Since the research results were determined, these results were validated and a sensitivity analysis was carried out. The validation of the research results was carried out in this way by using the calculated weights and the ranking of the alternatives was determined using other fuzzy methods in the form of Z-numbers. In this way, the ranking order of the alternatives obtained by the fuzzy CRADIS method were examined. After that, it was inspected as to how the weights of the criteria affected the final ranking of the alternatives. For this purpose, 60 scenarios were

used so that the importance of one criterion was reduced, and so that the criterion was used in the interval from 85% to 10%, while the other weights were changed. In this way, it was examined in what way the final ranking of the alternatives was affected.

### 2.1. Z-Numbers

Z-numbers represent an extension of the classic fuzzy number that provides the possibility of making decisions when there are additional uncertainties in decision making [31]. The main advantage of Z-numbers is that their strength lies in the use of uncertain information, which gives them an advantage over classical fuzzy numbers [32]. Due to the characteristic of including uncertainty in decision making, the concept of Z-numbers has been used with various methods of multicriteria decision making. Z-number represents an ordered pair of fuzzy numbers that appear as Z = (A, B), where A is a fuzzy number that represents the limit of variable X, while B is a fuzzy number that represents the dependence of fuzzy number A [33]. Generally, Z-numbers can be represented as: $\widetilde{Z} = \{(a_1, a_2, a_3; w_A), (b_1, b_2, b_3; w_B)\}$. The transformation of the Z-number into a classical fuzzy number is performed by applying the following steps [34]:

Step 1. Converting a B fuzzy number to a crisp number by applying defuzzification

$$\alpha = \frac{a_1 + a_2 + a_3}{3} \tag{1}$$

The problem with using this formula for defuzzification is that if the fuzzy numbers are proportionally far from each other, then the result of the defuzzification will always be $a_2$, so care must be taken that the values of the fuzzy numbers are not proportional, as in this case with B fuzzy numbers on which this formula is applied.

Step 2. Adding the weight of B fuzzy number to A fuzzy number

$$\widetilde{Z}^{\alpha} = \{\langle x, \mu_{A^{\alpha}}(x) \rangle | \mu_{A^{\alpha}}(x) = \alpha \mu_A(x)\} \tag{2}$$

Step 3. Converting a weighted Z-number to a regular fuzzy number

$$\widetilde{Z}' = \left\{ \langle x, \mu_{Z'}(x) \rangle | \mu_{Z'}(x) = \mu_A\left(\frac{x}{\sqrt{\alpha}}\right) \right\} \tag{3}$$

$$\widetilde{Z}' = \sqrt{a} \cdot \widetilde{A} = \left(\sqrt{a} \cdot a_1, \sqrt{a} \cdot a_2, \sqrt{a} \cdot a_3\right) \tag{4}$$

The use of $\sqrt{a}$ values was conducted in accordance with the research of Kang et al. [35], where the postulates of transforming Z-numbers into classic fuzzy numbers are set, which is why this term is used.

By using these steps, the Z-number is converted into an ordinary fuzzy number, and the operation with the used methods is performed in the same way as with ordinary fuzzy numbers. For more detailed information about the transformation of the Z-number into a fuzzy number, see the works of Kang et al. [33] and Božinić et al. [15]. Therefore, fuzzy LMAW and CRADIS methods are presented.

### 2.2. Z-Number–Fuzzy LMAW Method

The LMAW method was developed by Pamučar et al. [29]. This method enables determination of weights and ranking of alternatives [36], unlike other methods that serve only to determine weights or only to rank alternatives. The implementation of the complete fuzzy LMAW method is presented in [37]. Given that only part of the LMAW method is used, only the steps related to the calculation of the criterion weight coefficients are shown:

Step 1. Prioritization of criteria. In this step, each of the experts prioritizes the criteria $C = \{C_1, C_2, \ldots, C_n\}$ using two predefined scales, one for prioritizing the criteria and the other for defining the degree of confidence in the given prioritization. After surveying the experts, priority vectors $\widetilde{A}^e = \left(\widetilde{a}^e_{C1}, \widetilde{a}^e_{C2}, \ldots, \widetilde{a}^e_{Cn}\right)$ and confidence level vectors

$\widetilde{B}^e = \left( \widetilde{b}^e_{C1}, \widetilde{b}^e_{C2}, \ldots, \widetilde{b}^e_{Cn} \right)$ are defined, where $\widetilde{a}^e_{Cn}$ and $\widetilde{b}^e_{Cn}$, respectively, represent values from the fuzzy linguistic scales that expert $e$ ($1 \leq e \leq k$) assigned to criterion $n$.

Step 2. Converting Z-numbers into classical fuzzy numbers. This conversion is performed using Expressions (1)–(4). After executing the expression, new priority vectors are obtained separately for each expert $\widetilde{P}^e = \left( \widetilde{\gamma}^e_{C1}, \widetilde{\gamma}^e_{C2}, \ldots, \widetilde{\gamma}^e_{Cn} \right)$.

Step 3. Defining absolute fuzzy anti-ideal point $(\widetilde{\gamma}_{AIP})$. This is defined by the expert and this value represents a fuzzy number that is smaller than the smallest value from the set of all priority vectors.

Step 4. Defining the fuzzy vector of relations $(R^e = \left( \widetilde{\eta}^e_{C1}, \widetilde{\eta}^e_{C2}, \ldots, \widetilde{\eta}^e_{Cn} \right))$. By applying the Expression (5), the relation between the elements of the priority vector and the absolute anti-ideal point is determined $(\gamma_{AIP})$.

$$\widetilde{\mu}^e_{Cn} = \left( \frac{\widetilde{\gamma}^e_{Cn}}{\widetilde{\gamma}_{AIP}} \right) = \left( \frac{\gamma^{(l)e}_{Cn}}{\gamma^{(r)}_{AIP}}, \frac{\gamma^{(m)e}_{Cn}}{\gamma^{(m)}_{AIP}}, \frac{\gamma^{(r)e}_{Cn}}{\gamma^{(l)}_{AIP}} \right) \tag{5}$$

Step 5. Determining the vector of weighting coefficients, for each expert separately.

$$\widetilde{\omega}^e_j = \left( \frac{\ln\left( \widetilde{\mu}^e_{Cn} \right)}{\ln\left( \prod^n_{j=1} \widetilde{\mu}^e_{Cn} \right)} \right) = \left( \frac{\ln\left( \widetilde{\mu}^{(l)e}_{Cn} \right)}{\ln\left( \prod^n_{j=1} \widetilde{\mu}^{(r)e}_{Cn} \right)}, \frac{\ln\left( \widetilde{\mu}^{(m)e}_{Cn} \right)}{\ln\left( \prod^n_{j=1} \widetilde{\mu}^{(m)e}_{Cn} \right)}, \frac{\ln\left( \widetilde{\mu}^{(r)e}_{Cn} \right)}{\ln\left( \prod^n_{j=1} \widetilde{\mu}^{(l)e}_{Cn} \right)} \right) \tag{6}$$

Step 6. Calculation of aggregated fuzzy vectors of weight coefficient. The calculation of these weight coefficients is performed using the Bonferroni aggregator.

$$\widetilde{\omega}_j = \left( \frac{1}{k(k-1)} \sum^k_{\substack{i,j=1 \\ i \neq j}} \widetilde{\omega}^{(e)p}_i \widetilde{\omega}^{(e)q}_i \right)^{\frac{1}{p+q}} =$$

$$\left\{ \left( \frac{1}{k(k-1)} \sum^k_{\substack{i,j=1 \\ i \neq j}} \widetilde{\omega}^{(l_e)p}_i \widetilde{\omega}^{(l_e)q}_i \right)^{\frac{1}{p+q}}, \left( \frac{1}{k(k-1)} \sum^k_{\substack{i,j=1 \\ i \neq j}} \widetilde{\omega}^{(m_e)p}_i \widetilde{\omega}^{(m_e)q}_i \right)^{\frac{1}{p+q}}, \left( \frac{1}{k(k-1)} \sum^k_{\substack{i,j=1 \\ i \neq j}} \widetilde{\omega}^{(r_e)p}_i \widetilde{\omega}^{(r_e)q}_i \right)^{\frac{1}{p+q}} \right\} \tag{7}$$

Step 7. Calculation of final values of weighting coefficients. The final weights are obtained by defuzzification using the Expression (1).

*2.3. Z-Number–Fuzzy CRADIS Method*

The CRADIS method was developed by Puška et al. [38]. The method was developed to determine the distance of alternatives from ideal and anti-ideal solutions. Based on this distance, the alternatives are ranked. Since Z-numbers are used in this research, the steps Z-numbers–Fuzzy CRADIS method are used.

Step 1. Evaluation of alternatives by experts. In this step, experts evaluate alternatives $A = (A_1, A_2, \ldots, A_n)$ using two scales, one for evaluating alternatives and the other for defining the degree of certainty in a given evaluation. After that, the evaluation vectors of the alternatives $\widetilde{A}^e = \left( \widetilde{a}^e_{C1}, \widetilde{a}^e_{C2}, \ldots, \widetilde{a}^e_{Cn} \right)$ and the vector of the degree of certainty $\widetilde{B}^e = \left( \widetilde{b}^e_{C1}, \widetilde{b}^e_{C2}, \ldots, \widetilde{b}^e_{Cn} \right)$ are defined in the same way as was conducted for the Fuzzy LMAW method, only, instead of the criteria, the alternatives are evaluated. This is how Z-numbers are formed.

Step 2. Converting Z-numbers into classic fuzzy numbers. This conversion is performed using Expressions (1)–(4). After that, the steps of the fuzzy CRADIS method are applied [30]:

Step 3. Normalization of decision-making matrix.

For benefit criteria

$$\widetilde{n} = \left( n^l_{ij}, n^m_{ij}, n^u_{ij} \right) = \left( \frac{x^l_{id}}{x^u_{ij}}, \frac{x^l_{id}}{x^m_{ij}}, \frac{x^l_{id}}{x^l_{ij}} \right) \; if \; j \in C \tag{8}$$

For cost criteria

$$\widetilde{n} = \left( n_{ij}^l, n_{ij}^m, n_{ij}^u \right) = \left( \frac{x_{ij}^l}{x_{id}^u}, \frac{x_{ij}^m}{x_{id}^u}, \frac{x_{ij}^u}{x_{id}^u} \right) \ if \ j \in B \tag{9}$$

Step 4. Weighting of decision-making matrix. In this step, the values of the normalized decision matrix are multiplied by the corresponding weights.

$$\widetilde{v}_{ij} = \left( v_{ij}^l, v_{ij}^m, v_{ij}^u \right) = \widetilde{n}_j \times \widetilde{w}_j \tag{10}$$

Step 5. Determination of ideal and anti-ideal solutions. The ideal solution represents the highest value of the alternative in the weighted decision-making matrix, while the anti-ideal solution represents the lowest value of the alternative in the weighted decision-making matrix.

$$t_i = \max \widetilde{v}_{ij}, \ where \ \widetilde{v}_{ij} = \left( v_{ij}^l, v_{ij}^m, v_{ij}^u \right) \tag{11}$$

$$t_{ai} = \min \widetilde{v}_{ij}, \ where \ \widetilde{v}_{ij} = \left( v_{ij}^l, v_{ij}^m, v_{ij}^u \right) \tag{12}$$

Step 6. Calculation of deviations from ideal and anti-ideal solutions. Here, the deviation of the weighted data for the alternative in relation to the ideal and anti-ideal solution is calculated.

$$d^+ = t_i - \widetilde{v}_{ij} \tag{13}$$

$$d^- = \widetilde{v}_{ij} - t_{ai} \tag{14}$$

Step 7. Calculation of the evaluation of the deviation of individual alternatives from ideal and anti-ideal solutions. This step is performed in such a way that the deviation values for all alternatives in relation to the ideal and anti-ideal solution are added up.

$$s_i^+ = \sum_{j=1}^n d^+ \tag{15}$$

$$s_i^- = \sum_{j=1}^n d^- \tag{16}$$

Step 8. Defuzzification (converting fuzzy number to crisp number) deviations of alternatives from ideal and anti-ideal solutions.

$$s_{i\ def}^{\pm} = \frac{d_i^l + 4d_i^m + d_i^u}{6} \tag{17}$$

Step 9. Calculating the utility function. This step is performed in such a way that the deviation values of individual alternatives from ideal and anti-ideal solutions are compared with optimal alternatives. Optimal alternatives are the smallest values of deviation from the ideal solution; that is, the largest deviation values in relation to the anti-ideal solution.

$$K_i^+ = \frac{s_0^+}{s_i^+} \tag{18}$$

$$K_i^- = \frac{s_i^-}{s_0^-} \tag{19}$$

Step 10. Ranking of alternatives. The value of the fuzzy CRADIS method is determined by calculating the average value of the deviation of the alternatives from the utility function.

$$Q_i = \frac{K_i^+ + K_i^+}{2} \tag{20}$$

The ranking of alternatives is performed according to the value of the fuzzy CRADIS method.

## 3. Results

Before determining which of the suppliers had the best evaluation by the experts, it was necessary to first determine the weight of the criteria used. In order to determine the weight of the criteria and evaluate the suppliers, a nine-level value scale ranging from Absolutely low (AL) to Absolutely high (AH) was used (Table 2). This scale represents an A fuzzy number when converting into Z-numbers. In order to determine how confident decision makers are in their decision, it assesses the degree of certainty with a five-level scale ranging from Very small (VS) to Very high (VH). This value scale represents the B fuzzy number when converting Z-numbers.

**Table 2.** Linguistic values for evaluating the weight of the criteria and the alternative.

| Linguistic Value | Fuzzy Number A | Linguistic Value | Fuzzy Number B |
|---|---|---|---|
| Absolutely low (AL) | (1, 1, 1) | Very small (VS) | (0, 0, 0.2) |
| Very low (VL) | (1, 1.5, 2) | Small (S) | (0.1, 0.25, 0.4) |
| Low (L) | (1.5, 2, 2.5) | Medium (M) | (0.3, 0.5, 0.7) |
| Medium low (ML) | (2, 2.5, 3) | High (H) | (0.55, 0.75, 0.95) |
| Equal (E) | (2.5, 3, 3.5) | Very high (VH) | (0.8, 1, 1) |
| Medium high (MH) | (3, 3.5, 4) | | |
| High (H) | (3.5, 4, 4.5) | | |
| Very high (VH) | (4, 4.5, 5) | | |
| Absolutely high (AH) | (4.5, 5, 5) | | |

In order to determine the weight values, the experts evaluated the criteria (Table 3); in addition, since they had all the information about the criteria, they rated their level of security as High (H) and Very high (VH). This is because it represents their attitude about certain criteria and how important they are to them.

**Table 3.** Linguistic values when assessing the weight of criteria.

| | C1 | | C2 | | C3 | | C4 | | C5 | | C6 | | C7 | | C8 | | C9 | | C10 | |
|---|---|---|---|---|---|---|---|---|---|---|---|---|---|---|---|---|---|---|---|---|
| | A | B | A | B | A | B | A | B | A | B | A | B | A | B | A | B | A | B | A | B |
| Expert 1 (E1) | E | VH | H | VH | ML | H | EH | H | L | H | AH | VH | EH | H | MH | VH | L | H | E | H |
| Expert 2 (E2) | E | H | E | VH | H | H | ML | VH | E | VH | EH | H | AH | H | ML | H | E | VH | E | H |
| Expert 3 (E3) | MH | H | ML | H | L | VH | EH | H | AH | VH | EH | H | AH | VH | E | H | ML | H | L | H |

After the evaluations by the experts on the importance of individual criteria were obtained, the linguistic values were transformed into corresponding fuzzy numbers (Table 2) and the Z-numbers were formed. The next step was the transformation of Z-numbers into classical fuzzy numbers. The transformation was performed using Expressions (1)–(4) (Table 4).

**Table 4.** Transformed Z-numbers into classic fuzzy numbers.

| | C1 | C2 | C3 | C4 | C5 | ... | C9 | C10 |
|---|---|---|---|---|---|---|---|---|
| E1 | 2.42, 2.90, 3.38 | 3.38, 3.86, 4.35 | 1.73, 2.17, 2.60 | 3.46, 3.90, 4.33 | 1.30, 1.73, 2.17 | ... | 1.30, 1.73, 2.17 | 2.17, 2.60, 3.03 |
| E2 | 2.17, 2.60, 3.03 | 2.42, 2.90, 3.38 | 3.03, 3.46, 3.90 | 1.93, 2.42, 2.90 | 2.42, 2.90, 3.38 | ... | 2.42, 2.90, 3.38 | 2.17, 2.60, 3.03 |
| E3 | 2.60, 3.03, 3.46 | 1.73, 2.17, 2.60 | 1.45, 1.93, 2.42 | 3.46, 3.90, 4.33 | 4.35, 4.83, 4.83 | ... | 1.73, 2.17, 2.60 | 1.30, 1.73, 2.17 |

By transforming the Z-number into a classic fuzzy number, an initial (expert) decision matrix was obtained, which represents the first step in the implementation of the LMAW method. This was followed by step 3, which was the definition of the absolute fuzzy anti-ideal point ($\widetilde{\gamma}_{AIP}$). Then, the fuzzy relationship vectors were defined (Expression (5)). The next step was to determine the vector of weight coefficients for each expert separately (Expression (6)). Then the aggregate fuzzy vector of weight coefficients was calculated using Bonferroni aggregators (Expression (7)) (Table 5).

**Table 5.** Weight coefficients in the form of fuzzy numbers.

|  | | C1 | C2 | C3 | C4 | ... | C10 |
|---|---|---|---|---|---|---|---|
| | wj | 0.061, 0.095, 0.152 | 0.058, 0.096, 0.154 | 0.029, 0.071, 0.127 | 0.078, 0.115, 0.176 | ... | 0.023, 0.063, 0.116 |

The last step was the calculation of the final values of the criteria, which was performed by applying defuzzification (Expression (1)). Based on the obtained results, it can be concluded that the criteria Price (C7) and Quality (C6) received the highest value of weighting coefficients, while the criterion Innovation (C9) received the lowest value (Table 6). The difference between the largest and smallest values of the weight of the criteria that is, between criteria C7 and C9, was such that criterion C7 was 2.5 times higher than criterion C9. It should be emphasized that these results were obtained in accordance with the assessment of experts and based on their opinion on the importance of certain criteria.

**Table 6.** Weight coefficients of criteria.

|  | C1 | C2 | C3 | C4 | C5 | C6 | C7 | C8 | C9 | C10 |
|---|---|---|---|---|---|---|---|---|---|---|
| wj | 0.0990 | 0.0990 | 0.0733 | 0.1191 | 0.0919 | 0.1518 | 0.1558 | 0.0889 | 0.0606 | 0.0649 |

After the weights of the criteria were determined, the suppliers were ranked using the fuzzy CRADIS method. Before applying the steps of the fuzzy CRADIS method, it was necessary to transform the Z-numbers into classical fuzzy numbers. Before transforming the Z-numbers into classic fuzzy numbers and calculating the ranking of the alternatives, it was necessary to present the evaluations of the alternatives for the set criteria by the experts (Table 7).

Table 6 presents the assessment of alternatives by experts, where column A indicates the linguistic value for A fuzzy number representing the assessment of the criteria, while B indicates the linguistic value for B fuzzy number representing the reliability of the fuzzy number A, i.e., the certainty–uncertainty of the expert regarding the assessment of the value as an alternative. In order to transform Z-numbers, it is necessary to first transform the linguistic values into fuzzy numbers according to the membership function (Table 2). Then the steps of transformation of Z-numbers into classic fuzzy numbers (Expressions (1)–(4)) were performed. When Z-numbers are transformed into classic fuzzy numbers, these numbers are obtained for each expert separately. Since there were three experts in this research, in order to harmonize their opinions, the average value for each alternative was used. The average value was calculated using the arithmetic mean [39]. By applying the arithmetic mean, the same importance was given to experts in decision making. In this way, the initial decision matrix of the CRADIS method was obtained (Table 8).

**Table 7.** Evaluation of alternatives by experts.

| | Expert 1 | | | | | | | | | | | | | | | | | | | |
|---|---|---|---|---|---|---|---|---|---|---|---|---|---|---|---|---|---|---|---|---|
| | C1 | | C2 | | C3 | | C4 | | C5 | | C6 | | C7 | | C8 | | C9 | | C10 | |
| Alternative | A | B | A | B | A | B | A | B | A | B | A | B | A | B | A | B | A | B | A | B |
| A1 | MH | H | L | H | MH | H | MH | VH | E | H | AH | H | H | H | MH | H | ML | H | MH | H |
| A2 | EH | M | H | H | H | M | E | H | MH | H | H | H | EH | H | H | H | EH | H | EH | M |
| A3 | H | H | EH | M | EH | VH | AH | M | H | M | EH | VH | AH | M | EH | VH | H | VH | H | M |
| A4 | ML | H | E | H | L | H | L | VH | L | H | L | H | E | M | ML | M | MH | H | L | M |
| A5 | L | M | MH | H | ML | H | H | VH | EH | M | E | H | ML | H | E | H | E | H | E | H |
| A6 | E | H | ML | M | E | VH | ML | M | ML | M | ML | VH | MH | M | L | M | L | H | ML | M |

| | Expert 2 | | | | | | | | | | | | | | | | | | | |
|---|---|---|---|---|---|---|---|---|---|---|---|---|---|---|---|---|---|---|---|---|
| | C1 | | C2 | | C3 | | C4 | | C5 | | C6 | | C7 | | C8 | | C9 | | C10 | |
| Alternative | A | B | A | B | A | B | A | B | A | B | A | B | A | B | A | B | A | B | A | B |
| A1 | ML | M | E | S | H | H | ML | M | EH | VH | MH | H | MH | S | MH | M | EH | H | MH | VH |
| A2 | MH | H | H | M | E | H | H | S | MH | H | EH | H | AH | M | H | H | H | M | EH | VH |
| A3 | AH | H | AH | S | AH | VH | AH | S | AH | H | AH | H | EH | S | AH | H | AH | M | AH | VH |
| A4 | E | H | ML | S | ML | H | EH | H | L | H | ML | H | L | S | ML | H | E | M | E | H |
| A5 | H | M | MH | M | EH | H | MH | M | H | VH | H | M | H | S | EH | M | MH | H | H | H |
| A6 | EH | H | EH | S | L | H | E | M | ML | VH | E | H | ML | S | L | M | ML | H | ML | H |

| | Expert 3 | | | | | | | | | | | | | | | | | | | |
|---|---|---|---|---|---|---|---|---|---|---|---|---|---|---|---|---|---|---|---|---|
| | C1 | | C2 | | C3 | | C4 | | C5 | | C6 | | C7 | | C8 | | C9 | | C10 | |
| Alternative | A | B | A | B | A | B | A | B | A | B | A | B | A | B | A | B | A | B | A | B |
| A1 | MH | H | E | H | MH | VH | H | H | H | VH | H | H | E | H | E | H | E | M | MH | VH |
| A2 | EH | H | EH | H | AH | H | AH | VH | AH | VH | AH | H | EH | VH | EH | H | AH | H | EH | VH |
| A3 | E | M | H | M | EH | H | MH | H | MH | VH | E | M | AH | VH | AH | H | EH | H | AH | VH |
| A4 | L | M | ML | H | ML | VH | ML | H | E | H | ML | M | E | VH | ML | VH | ML | H | L | H |
| A5 | H | M | MH | H | H | H | EH | VH | EH | VH | EH | M | H | H | H | VH | H | M | H | VH |
| A6 | ML | H | L | H | E | VH | E | H | L | VH | MH | H | L | VH | L | H | L | H | ML | H |

**Table 8.** Initial fuzzy decision matrix.

| | C1 | C2 | C3 | C4 | C5 | C6 | C7 | . . . | C10 |
|---|---|---|---|---|---|---|---|---|---|
| A1 | 2.20, 2.61, 3.02 | 1.57, 1.94, 2.32 | 2.84, 3.29, 3.74 | 2.45, 2.87, 3.29 | 3.14, 3.60, 4.07 | 3.18, 3.61, 3.90 | 2.23, 2.60, 2.98 | . . . | 2.80, 3.26, 3.73 |
| A2 | 2.96, 3.37, 3.78 | 3.18, 3.61, 4.04 | 2.85, 3.25, 3.51 | 2.75, 3.14, 3.37 | 3.18, 3.63, 3.92 | 3.46, 3.90, 4.19 | 3.50, 3.93, 4.23 | . . . | 3.52, 3.96, 4.40 |
| A3 | 2.90, 3.31, 3.57 | 2.52, 2.84, 3.07 | 3.89, 4.36, 4.66 | 2.68, 3.02, 3.17 | 3.09, 3.51, 3.79 | 3.18, 3.60, 3.88 | 3.18, 3.54, 3.62 | . . . | 3.72, 4.16, 4.28 |
| A4 | 1.65, 2.06, 2.47 | 1.63, 2.00, 2.38 | 1.65, 2.10, 2.55 | 2.22, 2.66, 3.11 | 1.59, 2.02, 2.45 | 1.48, 1.89, 2.29 | 1.64, 2.01, 2.37 | . . . | 1.51, 1.91, 2.32 |
| A5 | 2.00, 2.36, 2.71 | 2.44, 2.85, 3.25 | 2.86, 3.31, 3.76 | 2.99, 3.41, 3.84 | 3.36, 3.80, 4.24 | 2.70, 3.11, 3.51 | 2.29, 2.68, 3.07 | . . . | 2.74, 3.18, 3.61 |
| A6 | 2.45, 2.89, 3.32 | 1.57, 1.92, 2.26 | 2.04, 2.51, 2.98 | 1.78, 2.16, 2.54 | 1.60, 2.04, 2.48 | 2.23, 2.68, 3.13 | 1.52, 1.89, 2.25 | . . . | 1.63, 2.03, 2.44 |

This was followed by step 3 of this method, which was data normalization. Since the questionnaire was designed in such a way that experts gave ratings for all criteria in the function of maximization, normalization for benefit criteria was applied (Expression (8)). The next step was weighting the normalized decision matrix (Expression (10)). This step was performed by multiplying the normalized values with the corresponding weights of the criteria. The next step was to determine the ideal and anti-ideal solutions (Expressions (11) and (12)). The ideal solution is the value of the fuzzy number that has the highest value in the weighted decision-making matrix, while the anti-ideal solution is the value of the fuzzy number that has the smallest value in the weighted decision-making matrix. Then followed the step of calculating the deviation from these solutions of all other data in the weighted decision matrix (Expressions (13) and (14)). The next step was to sum up the scores of deviations

from ideal and anti-ideal solutions ($s^+$, $s^-$) (Expressions (15) and (16)). This was followed by defuzzification of those values (Expression (17)). The next step was the calculation of the utility function in relation to the optimal alternatives (S0) for the ideal and anti-ideal solution (Expressions (18) and (19)). The last step was to calculate the average deviation of the alternatives from the degree of utility (Qi) (Expression (20)).

Based on the results obtained using the fuzzy CRADIS method, supplier A2 attained the best results, followed by supplier A3, while supplier A4 attained the worst results. According to these results, suppliers A2 and A3 should be the first choice for agricultural producers when purchasing raw and production materials for agricultural production. The obtained results (Table 9) show that these two suppliers have the highest values with the fuzzy CRADIS method (Q = 0.914 and Q = 0.885), while the other suppliers significantly deviate from these two suppliers. According to the opinion of the agricultural pharmacy, these suppliers are the most helpful for the application of green agricultural production. In this way, these results help individual agricultural producers to adapt to the demands of customers and markets.

**Table 9.** Deviation of alternatives from ideal solutions and final ranking order.

|  | $s^+$ | $s^-$ | Def $s^+$ | Def $s^-$ | $K_i^+$ | $K_i^-$ | $Q_i$ | RANK |
|---|---|---|---|---|---|---|---|---|
| A1 | (0.69, 0.74, 0.76) | (0.39, 0.42, 0.46) | 0.735 | 0.424 | 0.707 | 0.663 | 0.685 | 4 |
| A2 | (0.52, 0.57, 0.60) | (0.56, 0.60, 0.61) | 0.568 | 0.592 | 0.916 | 0.925 | 0.920 | 1 |
| A3 | (0.53, 0.59, 0.65) | (0.55, 0.58, 0.57) | 0.587 | 0.572 | 0.885 | 0.894 | 0.889 | 2 |
| A4 | (0.88, 0.93, 0.95) | (0.20, 0.23, 0.27) | 0.927 | 0.233 | 0.561 | 0.363 | 0.462 | 6 |
| A5 | (0.65, 0.71, 0.72) | (0.42, 0.46, 0.49) | 0.703 | 0.457 | 0.740 | 0.714 | 0.727 | 3 |
| A6 | (0.86, 0.92, 0.93) | (0.22, 0.25, 0.28) | 0.910 | 0.249 | 0.571 | 0.389 | 0.480 | 5 |
| $S_0$ | (0.47, 0.52, 0.55) | (0.60, 0.64, 0.66) | 0.520 | 0.640 |  |  |  |  |

### 3.1. Validation of Results

In order to confirm these results obtained by the fuzzy CRADIS method, the research results were validated. Validation represents a comparison of the results of several different methods of multicriteria analysis [40,41]. In this research, the results were compared with the following methods: fuzzy MARCOS (Measurement Alternatives and Ranking according to the COmpromise Solution), fuzzy WASPAS (weighted aggregated sum product assessment), fuzzy SAW (Simple Additive Weighting), fuzzy MABAC, fuzzy ARAS (Additive Ratio Assessment), fuzzy TOPSIS and fuzzy VIKOR (Višekriterijsko KOmpromisno Rangiranje). Each of these methods has its own specificities. The fuzzy MARCOS method uses the ranking of alternatives in relation to ideal and anti-ideal solutions, but, unlike the fuzzy CRADIS method, this method uses the determination of the degree of the utility function. The WASPAS method ranks alternatives based on a compromise between two methods: the Weighted Sum Model (WSM) and Weighted Product Model (WPM). The SAW method is the simplest of all the multicriteria analysis methods, and it only uses the aggregation of weighted normalized data. The MABAC method uses the comparison and determination of an approximate border area matrix in relation to the average value of the alternatives. The ARAS method ranks alternatives in relation to the degree of usefulness (optimal alternatives). The TOPSIS method uses the ranking of alternatives in relation to ideal and anti-ideal solutions, but the determination of these solutions and the implementation of this method differs from the MARCOS and CRADIS methods. The VIKOR method uses a compromise ranking of alternatives. Due to all these characteristics of the method and the results given, these methods may be different.

By implementing these methods, the results were identical to those obtained by the fuzzy CRADIS method (Figure 1). In this way, the results that suppliers A2 and A3 have the best characteristics were confirmed.

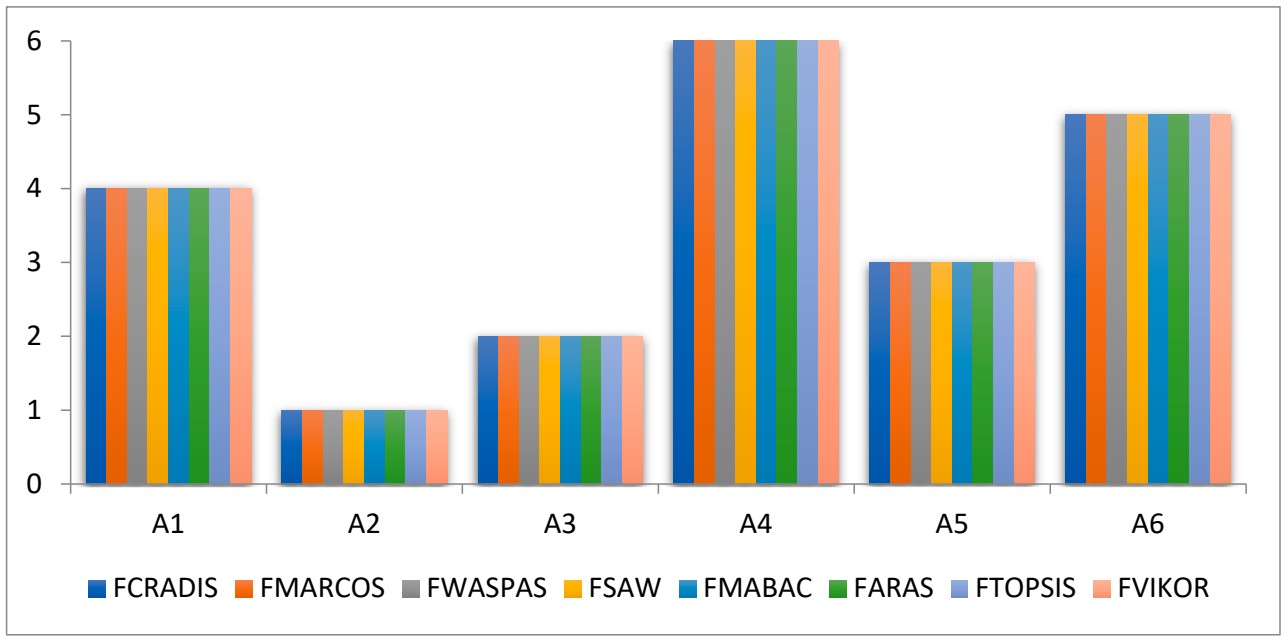

**Figure 1.** Validation of alternative ranking results.

*3.2. Sensitivity Analysis*

The next step in conducting this research was to perform a sensitivity analysis of the results. A sensitivity analysis was conducted in order to examine the influence of the criteria in the ranking of alternatives [42]. Since in practice there are several ways to conduct this analysis, the approach used by the authors Mešić et al. was chosen. [43]. Applying this way of carrying out the sensitivity analysis, the weights of one of the criteria are reduced in the ratio from 15% to 90%. In this way, it is observed how this weight reduction affects the final ranking of the alternatives. Since ten criteria were used in this research and the reduction was performed every 15%, i.e., six reductions were performed, a total of 60 scenarios were used (Table 10). In the first six scenarios, the weight of the first criterion was reduced by 15% until it reached 10% of the weight of that criterion, which was determined by the fuzzy LMAW method. In the other six scenarios, the weight reduction of the second criterion was performed, and so on until the weight reduction of the tenth criterion. In order to determine the weights of the other criteria where the weights do not change, the following formula was used

$$w_{n\beta} = (1 - w_{n\alpha})\frac{w_\beta}{(1 - w_n)} \qquad (21)$$

**Table 10.** Scenarios for carrying out sensitivity analysis.

| Scenario | C1 | C2 | C3 | C4 | C5 | C6 | C7 | C8 | C9 | C10 |
|---|---|---|---|---|---|---|---|---|---|---|
| S1 | 0.0784 | 0.0972 | 0.0766 | 0.1185 | 0.0916 | 0.1499 | 0.1536 | 0.0923 | 0.0671 | 0.0748 |
| S2 | 0.0665 | 0.0985 | 0.0776 | 0.1201 | 0.0928 | 0.1518 | 0.1556 | 0.0935 | 0.0680 | 0.0757 |
| S3 | 0.0538 | 0.0998 | 0.0786 | 0.1217 | 0.0941 | 0.1539 | 0.1577 | 0.0948 | 0.0689 | 0.0767 |
| S4 | 0.0403 | 0.1012 | 0.0797 | 0.1234 | 0.0954 | 0.1561 | 0.1599 | 0.0961 | 0.0699 | 0.0778 |
| S5 | 0.0260 | 0.1027 | 0.0809 | 0.1253 | 0.0968 | 0.1584 | 0.1623 | 0.0976 | 0.0710 | 0.0790 |
| S6 | 0.0107 | 0.1043 | 0.0822 | 0.1272 | 0.0983 | 0.1609 | 0.1648 | 0.0991 | 0.0721 | 0.0802 |
| S7 | 0.1006 | 0.0758 | 0.0765 | 0.1184 | 0.0915 | 0.1498 | 0.1534 | 0.0922 | 0.0671 | 0.0747 |
| ⋮ | ⋮ | ⋮ | ⋮ | ⋮ | ⋮ | ⋮ | ⋮ | ⋮ | ⋮ | ⋮ |
| S59 | 0.1043 | 0.1006 | 0.0793 | 0.1227 | 0.0948 | 0.1552 | 0.1590 | 0.0956 | 0.0695 | 0.0190 |
| S60 | 0.1055 | 0.1018 | 0.0802 | 0.1241 | 0.0959 | 0.1570 | 0.1608 | 0.0967 | 0.0703 | 0.0078 |

By carrying out this procedure, the weight of the criteria is first reduced, and then the calculation is made of how much the other criteria lack so that their total value is 1, and then the values of the other criteria are adjusted for the difference of the reduction of the criterion that is being reduced.

During the implementation of these 60 scenarios in the sensitivity analysis, such results were obtained that there was no change in the ranking of the four first-ranked alternatives (Figure 2). Only in five scenarios was there a difference in the ranking of alternatives A6 and alternative A4, where alternative A4 was ranked better than alternative A6. This analysis only confirms the obtained results of this research.

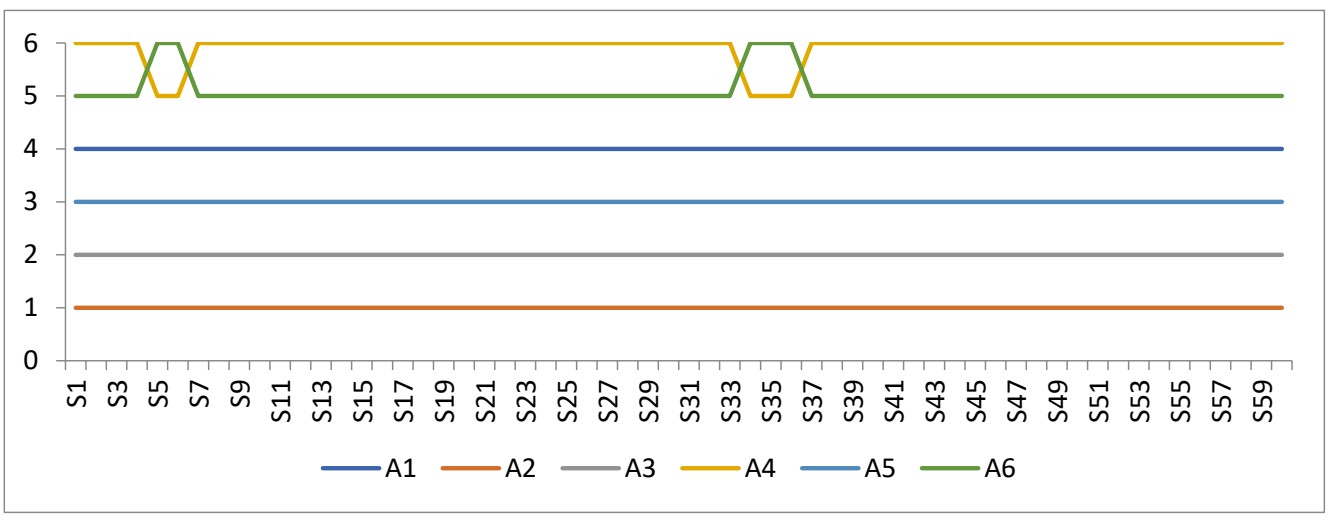

**Figure 2.** Results of the sensitivity analysis.

## 4. Discussion

When starting any production, it is necessary to use various raw and production materials. The use of raw materials largely determines what the final product will be. In agriculture, it is characteristic that the market demands that products should be as environmentally friendly as possible and harmless to the human body [1]. Therefore, it is necessary to choose raw and production materials that enable green agricultural production. In order to implement this agricultural production, various criteria are used in the selection of suppliers. In order to achieve this in practice, GSS is used. With this choice, care is taken to reduce the negative impact on the environment and to maximize the economic effects [10]. In this research, GSS was used for agricultural production purposes. In doing so, ten criteria were used, which were evenly divided into ecological and economic criteria. In this way, the same importance was given to these criteria. These criteria were chosen together with the experts, and they are, in their opinion, the most important for the GSS.

In order to evaluate suppliers of raw and production materials, an example based on agricultural companies for distribution was used. They have a great impact on agricultural production in Bosnia and Herzegovina [25]. In addition, agricultural companies for distribution buy these raw materials in bulk from producers, so they know more about them than agricultural producers themselves. This research was conducted on the example of the agricultural company "Agromarket" in Bijeljina. The reason for this is that it has three distribution centres through which it sells raw and production materials for agriculture throughout the Republic of Srpska and Bosnia and Herzegovina. In order to select suppliers, expert decision making was used. However, the use of expert opinion entails the fact that some of the experts will evaluate individual suppliers too subjectively [44]. That is why Z-numbers were used. When evaluating the criteria and alternatives, the experts were asked how confident they were in their assessment because, when making a decision, they cannot have all the information about the suppliers and their raw and production materials

in order to realistically evaluate the criteria and alternatives. Therefore, in these and similar decision-making processes, there is uncertainty, so Z-numbers provide good support.

Z-numbers have been used in many studies in the selection of suppliers in combination with methods of multicriteria analysis [19,21,23]. By applying Z-numbers, an effort was made to reduce uncertainty in supplier evaluation. Experts were faced with the problem that they did not have all the information about their suppliers, so this lack of information was compensated for by using Z-numbers. In some cases, these numbers were also used in the case of ambiguity when choosing a supplier [19]. Based on this, it can be concluded that the application of Z-numbers is large, although, in practice, it is not used as widely as it should be. Z-numbers should be used in all decision making wherever there is some uncertainty in the estimation, where relative probability theory [18], a genetic algorithm [22] and other similar applications can be used. In this research, the use of Z-numbers was combined with the fuzzy methods LMAW and CRADIS, which are newer multicriteria methods. Based on this, it is possible to combine these Z-numbers with other fuzzy methods very easily.

When evaluating the criteria, the experts gave a linguistic value of nine levels ranging from Absolutely low (AL) to Absolutely high (AH). However, when evaluating criteria, the opinion of experts is sought as to how important a certain criterion is in their opinion. Therefore, they rated their confidence in this assessment as High and Very high because the experts were very confident in their opinion. By first applying the conversion of Z-numbers into classic fuzzy numbers, and then the calculation using the fuzzy LMAW method, such values were obtained that, according to experts, the most important criteria are price and quality. Similar results were obtained by Matić et al. [2], with the fact that, in their research, the quality criterion was given the greatest weight, followed by the price/cost criterion. In contrast to this conducted research, Tirkolaee et al. [9] confirmed in their research that price is the most important criterion for GSS. This is due to the fact that the price is the key economic indicator of agricultural production [45]. Thus, according to the opinion of these experts, it is necessary to acquire raw and production materials of excellent quality and at affordable prices. Similar results were obtained by Adeel Jafar et al. [46] who concluded that agricultural products should be of good quality. In addition to these two economic criteria, there are three ecological criteria, so it is necessary to take ecological criteria into account when selecting green suppliers. This means that raw and production materials procured from GSS must also be environmentally acceptable, and not only of good quality and affordable prices.

During the evaluation of the suppliers, there was a greater uncertainty in the evaluation of the experts, which could be seen in the value of the B fuzzy number. This is because, although the suppliers are known, they do not have all the information about them. This uncertainty lowers the ratings of those suppliers where experts are more uncertain. The reason for this is the conversion of Z-numbers into classic fuzzy numbers presented by Kang et al. [33]. If the value of the B fuzzy number is smaller, that is, if the expert is more uncertain, he will choose a smaller value, and during the defuzzification of these fuzzy numbers, the number of crips will be smaller. If that crips number is smaller, when multiplying with A fuzzy number, a smaller value of the classic fuzzy number will be obtained. In this way, that supplier will be ranked lower. Therefore, it is necessary to have equal information about all suppliers in order for ranking to be as realistic as possible.

Based on expert evaluations and ranking using the fuzzy CRADIS method, results showed that suppliers A2 and A3 have the best indicators. The values obtained using the fuzzy CRADIS method showed that these six suppliers were gathered into three groups, namely, suppliers A2 and A3 who showed the best results, then suppliers A5 and A1 who showed medium results, and suppliers A6 and A4 who showed the worst results. The same results were obtained using other fuzzy methods. Based on that, the results obtained using the fuzzy CRADIS method were confirmed. The sensitivity analysis matched the gathering of suppliers into three groups, so the ranking order in the five scenarios was changed in the group of suppliers that showed the worst results, namely, suppliers A6 and A4. When

changing the importance of the criteria green product (C1) and the quality criteria (C6) and reducing their values, supplier A4 was ranked better than supplier A6. This shows that these two suppliers have equal evaluations by experts, and that the biggest difference is in these two criteria. This is why supplier A4 was ranked fifth. What the sensitivity analysis also shows is that the other four suppliers did not change the ranking order. In this way, the results of this research were confirmed and those suppliers who can help with green agricultural production were selected.

## 5. Conclusions

In this paper, GSS was used in agricultural production. The specificity of this research is the use of Z-numbers that reduce uncertainty in decision making. The implementation of this research has shown how agricultural production can be improved by applying GSS because applying ecological and economic criteria gives a broader picture of suppliers and enables, with the choice of the right supplier, the application of green agricultural production. In addition, the application of Z-numbers reduces uncertainty in decision making.

In this research, expert decision making was applied for GSS. The experts evaluated the criteria and alternatives and applied the fuzzy LMAW and CRADIS methods when using Z-numbers. The obtained results of using fuzzy LMAW show that the most important criteria when choosing a supplier in agricultural production are quality and price. When ranking suppliers using the fuzzy CRADIS method, the best suppliers are A2 and A3, that is, they show the best results. It is necessary to establish partnership relations with these suppliers because their raw and production materials contribute to the development of green agricultural production, which is in accordance with the requirements of customers and the market.

Based on the conducted research and the obtained results, it can be concluded that the application of Z-numbers is not so complex that it should not be used. In future research, Z-numbers should be used in other research where there is uncertainty in decision making. Due to the prevalence of uncertainty in decision making, Z-numbers can be used in any field, so they should be used more often in future research.

The transformation of Z-numbers into classic fuzzy numbers is not complex; the only thing that is complex is for the experts to evaluate the criteria and alternatives because they must also give an evaluation of how confident they are in their evaluation. That part can be considered a disadvantage of using Z-numbers, but in this way the uncertainty in decision making is solved. Using Z-numbers with fuzzy LMAW and CRADIS methods showed that they can be used with different methods; so, in future research, Z-numbers can be used with various methods of multicriteria analysis. In future papers, Z-numbers should be used in combination with other methods as their simplicity in use has been proven.

**Author Contributions:** Conceptualization, A.P., D.B. and M.N.; methodology, A.P. and D.B. software, A.P. and D.B.; validation, M.N. and M.J.; formal analysis A.P. and D.B.; investigation, M.N.; resources, M.N.; data curation, A.P. and M.N.; writing—original draft preparation, A.P., D.B. and M.N.; writing—review and editing, A.P., D.B. and M.N.; visualization, A.P. and D.B.; supervision, M.N.; project administration, M.J.; funding acquisition, M.N. All authors have read and agreed to the published version of the manuscript.

**Funding:** This research received no external funding.

**Institutional Review Board Statement:** Not applicable.

**Informed Consent Statement:** Not applicable.

**Data Availability Statement:** Not applicable.

**Acknowledgments:** Not applicable.

**Conflicts of Interest:** The authors declare no conflict of interest.

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
