# Peer review of "Green Supplier Selection in an Uncertain Environment in Agriculture Using a Hybrid MCDM Model: Z-Numbers–Fuzzy LMAW–Fuzzy CRADIS Model"

_axioms, doi:10.3390/axioms11090427_

Round 1
Reviewer 1 Report
(1) Novelty of research needs to be discussed.
(2) carefully proof editing the language of the paper.
(3) providing complete reference details.
(4) numbering all the main and sub-sections to improve the flow of the paper.
(5) duly following the format of Axioms for the whole manuscript including the references.
(6) The manuscript's result needs to be compared with the similar papers in the conclusion section
(7) More future research needs to be added to the paper.
Author Response
(1) Novelty of research needs to be discussed.
Added this item
(2) carefully proof editing the language of the paper.
The paper will be thoroughly proofread once the reviews are complete.
(3) providing complete reference details.
References are edited in the paper
(4) numbering all the main and sub-sections to improve the flow of the paper.
Main and sub-sections are numbered
(5) duly following the format of Axioms for the whole manuscript including the references.
References have been corrected in accordance with the instructions
(6) The manuscript's result needs to be compared with the similar papers in the conclusion section
In the discussion, the results were compared with similar papers
(7) More future research needs to be added to the paper.
Directions for future research are provided
Reviewer 2 Report
This manuscript proposes a hybrid multi-criteria decision-making (MCDM) approach for the green supplied selection (GSS) process in uncertain agricultural supply chains. The proposed hybrid MCDM approach combines a Logarithm Methodology of Additive Weights (LMAW) using Z numbers with the Compromise Ranking of Alternatives from Distance to Ideal Solution (CRADIS) approach. The developed approach is applied to a case study in Bosnia and Herzegovina. This study suggests that the proposed approach can reduce the uncertainty in decision making as a result of utilizing Z numbers; which in turn may have some complexity in its application as the experts have to assess their confidence in their evaluations.
This research provides some insights into the adoption of Z numbers in MCDM models. However, its contribution is marginal since it is not benchmarked against other approaches that can be used to address uncertainty in the experts’ evaluations. The literature review of this manuscript needs to be strengthened by adding more references of how uncertainties in MCDM are dealt with and focusing on two or more alternative approaches that can be used to benchmark the proposed approach in the considered case study.
Other that the above concern, the manuscript contains a few typos that required careful proofreading. Some formatting issues need to be resolved such as the locations of the tables so that they do not break between pages. Specifically, the rows in Table 1 need to be properly aligned.
Author Response
This manuscript proposes a hybrid multi-criteria decision-making (MCDM) approach for the green supplied selection (GSS) process in uncertain agricultural supply chains. The proposed hybrid MCDM approach combines a Logarithm Methodology of Additive Weights (LMAW) using Z numbers with the Compromise Ranking of Alternatives from Distance to Ideal Solution (CRADIS) approach. The developed approach is applied to a case study in Bosnia and Herzegovina. This study suggests that the proposed approach can reduce the uncertainty in decision making as a result of utilizing Z numbers; which in turn may have some complexity in its application as the experts have to assess their confidence in their evaluations.
This research provides some insights into the adoption of Z numbers in MCDM models. However, its contribution is marginal since it is not benchmarked against other approaches that can be used to address uncertainty in the experts’ evaluations. The literature review of this manuscript needs to be strengthened by adding more references of how uncertainties in MCDM are dealt with and focusing on two or more alternative approaches that can be used to benchmark the proposed approach in the considered case study.
Other that the above concern, the manuscript contains a few typos that required careful proofreading. Some formatting issues need to be resolved such as the locations of the tables so that they do not break between pages. Specifically, the rows in Table 1 need to be properly aligned.
When the paper corrections are finished, the tables are now on one page. Detailed proofreading will be done once the proofreading of the paper is complete.
Reviewer 3 Report
Authors investigate a problem of " Title: Green supplier selection in an uncertain environment in agriculture using a hybrid MCDM model: Z-numbers fuzzy LMAW-fuzzy CRADIS model ".
The work aims to illustrate the selection of the right supplier (green supplier - GSS) who can provide lower operating costs, higher product quality with minimal environmental impact. GSS is considered a problem that is solved by applying group multi-criteria decision-making. The authors use methods as mathematical methods: Z-TOPSIS (Z - Technique for Order of Preference by Similarity to Ideal Solution); Z-MABAC (Z - Multi-Attributive Border Approximation Area Comparison); BWM (Best Worst Method); и TODIM (Tomada de Decisão Iterativa Multicritério); LMAW (Logarithm Methodology of Additive Weights) and fuzzy CRADIS (Compromise Ranking of Alternatives from Distance to Ideal Solution) et al.
Let's pass to mathematics.
Section 2 presents the Methodology and methods, which includes three subsections: 2.1. Z-numbers; 2.2. Z number - Fuzzy LMAW method; 2.3. Z-number-fuzzy CRADIS method.
Every step of the methodology is not theoretically justified. For example, Section 2.2 in step 1 uses the formula for determining the average number: ? = (?1+?2+?3 )/3 (1). It is quite clear that the result will be ?2, and table 2 of the third section confirms this.
Step 3 uses the square root of the √α. Question: Why not the third or tenth degree?
Step 6 uses the sum of the indicators. Question: Why isn't the product of metrics used?
As a result, numbers are obtained, the value of which does not make any sense - neither economic nor engineering. The methodology is based on the concept of "Weighting coefficients". But the author do not understand the influence of "weighting factors" on the result of the decision (as well as the authors of literary references).
The author suggest that the weighting factors equalize the criteria for making decisions. The more weight, the more important (priority) the corresponding criterion.
For example, from Table 6 Weight coefficients of criteria (Criteria: C9 0.0606; C6 0.1518) follows: w1 weight=0.0606 rounded w1=0.06
w2 weight=0.1518 rounded w2=0.15
follows that w2 =0.15 is more important (priority) w1=0.06.
Let's analyse the result mathematically.
The value of the first indicator W1 indicates "a" and the value of the second indicator W2 indicates "b." The work assumes that the weights equalize their effect on the task.
Then the relation is carried out
w1*a = w2*b or
0.06*a=0.15*b
From here
a=0.15*b/0.06 or
a=2.5*b.
As a result, we learn that the W1 criterion is 2.5 times more effective than the W2 criterion, i.e. the result with accuracy vice versa.
In work https://rdcu.be/bhZ8i in the application
I conducted a theoretical analysis of the use of weights and showed that they give quite the wrong result that the author puts into them.
Therefore, all further arithmetic actions will lead to the answer that the authors need.
Thus, in general, the review of the article is negative.
Recommendations.
At the same time, I believe that the article should be printed in your journal (together with the review). The volume of work shows that the authors worked on the article. In science (and mathematics especially), a negative result is also a result. It is quite possible that I did not see something that will interest readers.
Therefore, I recommend the work to be published together with the review.
Author Response
Authors investigate a problem of " Title: Green supplier selection in an uncertain environment in agriculture using a hybrid MCDM model: Z-numbers fuzzy LMAW-fuzzy CRADIS model ".
The work aims to illustrate the selection of the right supplier (green supplier - GSS) who can provide lower operating costs, higher product quality with minimal environmental impact. GSS is considered a problem that is solved by applying group multi-criteria decision-making. The authors use methods as mathematical methods: Z-TOPSIS (Z - Technique for Order of Preference by Similarity to Ideal Solution); Z-MABAC (Z - Multi-Attributive Border Approximation Area Comparison); BWM (Best Worst Method); и TODIM (Tomada de Decisão Iterativa Multicritério); LMAW (Logarithm Methodology of Additive Weights) and fuzzy CRADIS (Compromise Ranking of Alternatives from Distance to Ideal Solution) et al.
Let's pass to mathematics.
Section 2 presents the Methodology and methods, which includes three subsections: 2.1. Z-numbers; 2.2. Z number - Fuzzy LMAW method; 2.3. Z-number-fuzzy CRADIS method.
Every step of the methodology is not theoretically justified. For example, Section 2.2 in step 1 uses the formula for determining the average number: ? = (?1+?2+?3 )/3 (1). It is quite clear that the result will be ?2, and table 2 of the third section confirms this.
It is explained when this term should not be used.
Step 3 uses the square root of the √α. Question: Why not the third or tenth degree?
What this is used for is explained
Step 6 uses the sum of the indicators. Question: Why isn't the product of metrics used?
As a result, numbers are obtained, the value of which does not make any sense - neither economic nor engineering. The methodology is based on the concept of "Weighting coefficients". But the author do not understand the influence of "weighting factors" on the result of the decision (as well as the authors of literary references).
The author suggest that the weighting factors equalize the criteria for making decisions. The more weight, the more important (priority) the corresponding criterion.
For example, from Table 6 Weight coefficients of criteria (Criteria: C9 0.0606; C6 0.1518) follows: w1 weight=0.0606 rounded w1=0.06
w2 weight=0.1518 rounded w2=0.15
follows that w2 =0.15 is more important (priority) w1=0.06.
Let's analyse the result mathematically.
The value of the first indicator W1 indicates "a" and the value of the second indicator W2 indicates "b." The work assumes that the weights equalize their effect on the task.
Then the relation is carried out
w1*a = w2*b or
0.06*a=0.15*b
From here
a=0.15*b/0.06 or
a=2.5*b.
As a result, we learn that the W1 criterion is 2.5 times more effective than the W2 criterion, i.e. the result with accuracy vice versa.
It is explained in more detail why one criterion is 2.5 times more important than another criterion.
In work https://rdcu.be/bhZ8i in the application
I conducted a theoretical analysis of the use of weights and showed that they give quite the wrong result that the author puts into them.
Therefore, all further arithmetic actions will lead to the answer that the authors need.
Thus, in general, the review of the article is negative.
Recommendations.
At the same time, I believe that the article should be printed in your journal (together with the review). The volume of work shows that the authors worked on the article. In science (and mathematics especially), a negative result is also a result. It is quite possible that I did not see something that will interest readers.
Therefore, I recommend the work to be published together with the review.
An attempt has been made to correct everything you found objectionable in the paper.
Round 2
Reviewer 2 Report
This manscript is acceptable.